

# Towards a manual-free labelling approach for deep learning-based ice floe instance segmentation in airborne and high-resolution optical satellite images

Qin Zhang[1] and Nick Hughes[1]

[1]Norwegian Meteorological Institute, Norwegian Ice Service, Kirkegårdsveien 60, P.O. Box 6314 Langnes, Tromsø 9293, Norway

**Correspondence:** Qin Zhang (qinz@met.no, chin.qz.chang@gmail.com)

**Abstract.** Floe size distribution (FSD) has become a parameter of great interest in observations of sea ice because of its importance in affecting climate change, marine ecosystems, and human activities in the polar ocean. The sizes of ice floes can range from less than a square metre to hundreds of square kilometres, so the most effective way to monitor FSD in the ice-covered regions is to apply image processing techniques to airborne and satellite remote sensing data. The segmentation of individual ice floes is crucial for obtaining FSD from remotely sensed images, and it is a challenge to separate floes that appear to be connected. Although deep learning (DL) networks have achieved great success in image processing, they still have limitations in this application. A key reason is the lack of sufficient labelled data, which is costly and time-consuming to produce. In order to alleviate this issue, we use classical image processing techniques to achieve a manual-label free ice floe image annotation, which is further used to train DL models for fast and adaptive individual ice floe segmentation, especially for separating visibly connected floes. A post-processing algorithm is also proposed in our work to refine the segmentation. Our approach has been applied to both airborne and high-resolution optical (HRO) satellite images, and successfully derived FSD at local and global scales.

## 1 Introduction

Determining the characteristics of sea ice is critical to the understanding of physical processes in the polar regions and climate change globally (Notz and SIMIP Community, 2020). Although sea ice concentration (SIC) and sea ice thickness (SIT) are widely used parameters, the size and shape distributions of individual pieces of sea ice, floes, are also important for determining the SIC (Nose et al., 2020), processes including sea ice melt (Horvat and Tziperman, 2018), propagation of ocean waves in the ice pack (Squire et al., 1995), development and maintenance of the upper ocean mixed layer (Manucharyan and Thompson, 2017), and also play an important role in human activities, such as maritime navigation and offshore operations in ice-covered regions (Marchenko, 2012; Mironov, 2012). Image data from various sources are rich in environmental information, and from which many floe parameters can be extracted to determine a floe size distribution (FSD) (Rothrock and Thorndike, 1984). The extraction of individual ice floes is crucial for determining FSD and other floe characteristics from images, and the separation of connected floes has always been a challenge. The existing methods for extracting individual ice floes and estimating FSD



from images are mainly based on classical image processing methods. A simple approach is to define an ice-water segmen-
tation threshold to extract floes and then apply manual edge corrections when the threshold performs poorly (Toyota et al.,
2006, 2011, 2016). Watershed transform has been adopted to detach connected floes, but excessive over-segmentation is an
ineluctable problem when using this method (Blunt et al., 2012; Zhang et al., 2013). Morphological operations can be used with
different improvements to determine individual ice floes, but the methods operate directly on binarized floe images and thus
cannot separate out the floes that had no or few gaps with any surrounding floes after binarization (Banfield, 1991; Banfield
and Raftery, 1992; Soh et al., 1998; Steer et al., 2008; Wang et al., 2016). The gradient vector flow (GVF) snake was used
by (Zhang and Skjetne, 2015) for detecting weak floe boundaries and has achieved excellent results in segmenting individual
floes from MIZ images, in which a large amount of floes of similar shapes and sizes are connected to each other. However,
this method is not time-effective and may not work well with floe images other than MIZ images, especially the larger scale
images such as satellite imagery (Zhang, 2020; Zhang and Skjetne, 2018). These classical image processing based methods
suffered from segmentation problems, and are more or less limited by the need for manual intervention in processing individual
images. Due to the vast volumes of image data now being collected by Earth Observation programmes such as Copernicus,
manual intervention is undesirable due to its inefficiency. Autonomous, trustworthy, and time-efficient methods thus need to
be developed.

Deep learning (DL) methods have nowadays proven to deliver superior accuracy in a wide range of image processing
applications. Pixel-based DL methods are able to map complex features at the pixel level from an image in an automated
process, and they have also been applied in extracting all ice pixels from images consisting of a mixture of sea ice and water
(Khaleghian et al., 2021; Gonçalves and Lynch, 2021; Zhang et al., 2021). However, most of these studies grouped the pixels
belonging to different ice regions or floes into the same class (i.e. the class of ice), and did not contribute to the identification
of individual ice floes, which is an instance segmentation problem that requires multiple ice floes to be treated as distinct
individual instances. Only few studies have employed DL methods to identify individual ice floes. A semantic segmentation
model, ResUNet, was used in (Nagi et al., 2021) to segment individual floes that were far apart from each other, and then
used ConvCRF to refine the segmentation results. This method, however, simply divided an image into two classes of ice
and background, and was unable to separate connected floes. (Cai et al., accepted) has adopted and compared two state-of-art
(SoA) DL instance segmentation models, Mask R-CNN (He et al., 2017) and YOLACT (Bolya et al., 2019), for identifying
individual model floes in an indoor ice tank. Because these models rely heavily on their own object detectors to produce
instance segmentation results, neither model could fully detect every floe appeared in the image, resulting in the loss of floes,
and the segmentation accuracy was usually not high.

Training a DL model requires a sufficiently annotated dataset. However, data labelling usually involves a lot of manual
work and is expensive and time-consuming, which limits the application of DL methods to extracting individual ice floes (Jing
and Tian, 2021; Zhou, 2017; Chai et al., 2020). In order to minimise the manual labelling effort required from the domain
experts, we use classical image processing method to enable a manual-label free annotation of the dataset and automatically
generate pseudo ground truth. We then apply DL semantic segmentation method, which assigns every pixel in an image to
defined classes, to address floe instance segmentation problem, and propose a post-processing algorithm to refine the model



outputs. The application of our approach to derive FSDs from local-scale airborne imagery and global-scale satellite imagery demonstrates the effectiveness of our approach.

## 2 Dataset

### 2.1 Local-scale Airborne data

The local-scale image data are marginal ice zone (MIZ) images mainly from the Oden Arctic Technology Research Cruise 2015 (OATRC'15) expedition. OATRC'15 was conducted by the Norwegian University of Science and Technology (NTNU) in collaboration with the Swedish Polar Research Secretariat (SPRS) in September 2015 (Lubbad et al., 2018). Two icebreakers, *Oden* and *Frej*, were employed during this research cruise into the Arctic Ocean north of Svalbard. Among many research activities, a helicopter flight mission was accomplished when *Oden* was transiting in the MIZ during which an optical camera was mounted on the helicopter, as seen in Fig. 1. This enabled the acquisition of a large number of very high resolution images of sea ice. The resolution of the collected MIZ image depends on helicopter's flight altitude, and was estimated as 0.22 m on average. The specifications of the camera can be found in Table 1 (Zhang and Skjetne, 2018).

**Table 1.** Visible spectrum camera specifications.

| Lens type | Fujion 35mm |
|---|---|
| Focal length | 28.5 mm |
| Dimensions | $5568 \times 3132$ |
| Sampling frequency | 0.1 Hz |

In addition to the MIZ images from OATRC'15, we also used another three airborne images obtained from the remote sensing UAV (unmanned aerial vehicle) mission over the MIZ performed by the Northern Research Institute (NORUT) at Ny-Ålesund, Svalbard ($78°55'N$ $11°56'E$), in early May, 2011. The details of this UAV mission and the specifications of the collected image data can be found in (Zhang et al., 2012).

### 2.2 Global-scale Satellite data

The global-scale high-resolution optical (HRO) satellite imagery data are the freely-available data from the Sentinel-2 mission of the European Copernicus programme. The two Sentinel-2 satellites carry a multispectral instrument (MSI) that provides images consisting of 13 spectral bands: four bands at 10 m resolution covering visible and near-infrared (VNIR) frequencies, six at 20 m covering red edge and short-wave infrared (SWIR), and three at 60 m for atmospheric correction (Drusch et al., 2012). Fig. 2 displays the locations of the four Sentinel-2 Level-1 images that were acquired over the Belgica Bank area offshore of north-east Greenland during May and June 2021, and in this article we refer to them as S2-1, S2-2, S2-3, and S2-4. Table 2 lists and identifies the filenames used (S2n). More details for each S2 image can be found by searching for their filenames on the Copernicus Open Access Hub (Cop).



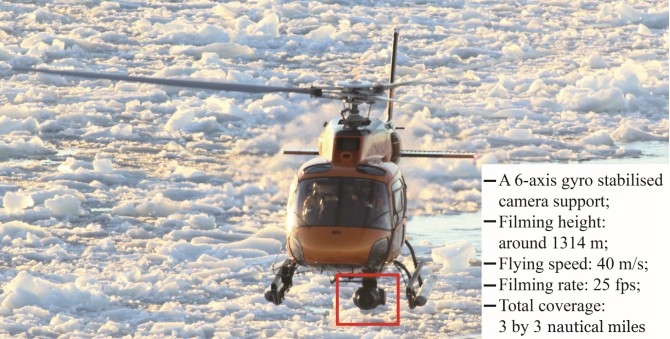

(a) Helicopter camera.

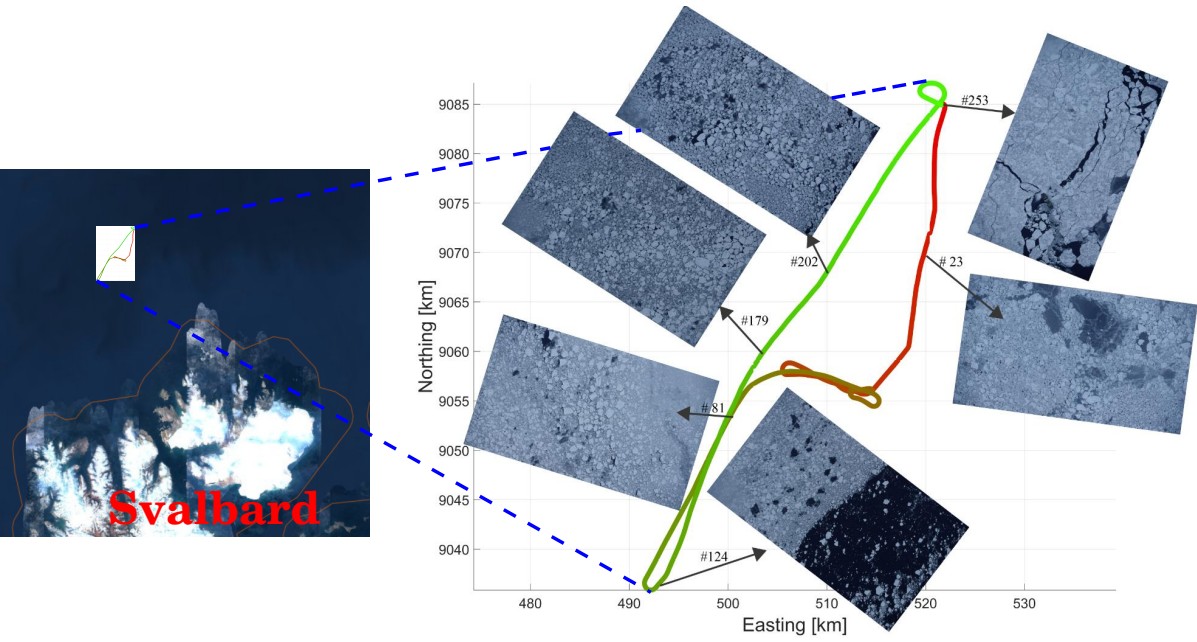

(b) Helicopter's flying route, starting from red and gradually changing into green. Source: (Lubbad et al., 2018).

**Figure 1.** Helicopter flight mission. (a) Helicopter showing location of the camera system; (b) Flight route.

**Table 2.** Sentinel-2 image data.

| Item | Product ID |
|------|------------|
| S2-1 | S2A_MSIL1C_20210527T144921_N0300_R082_T28XEN_20210527T165730 |
| S2-2 | S2B_MSIL1C_20210528T150759_N0300_R025_T29XMJ_20210528T171359 |
| S2-3 | S2A_MSIL1C_20210617T141951_N0300_R096_T28XEL_20210617T162322 |
| S2-4 | S2B_MSIL1C_20210620T133729_N0300_R067_T29XNG_20210620T154321 |



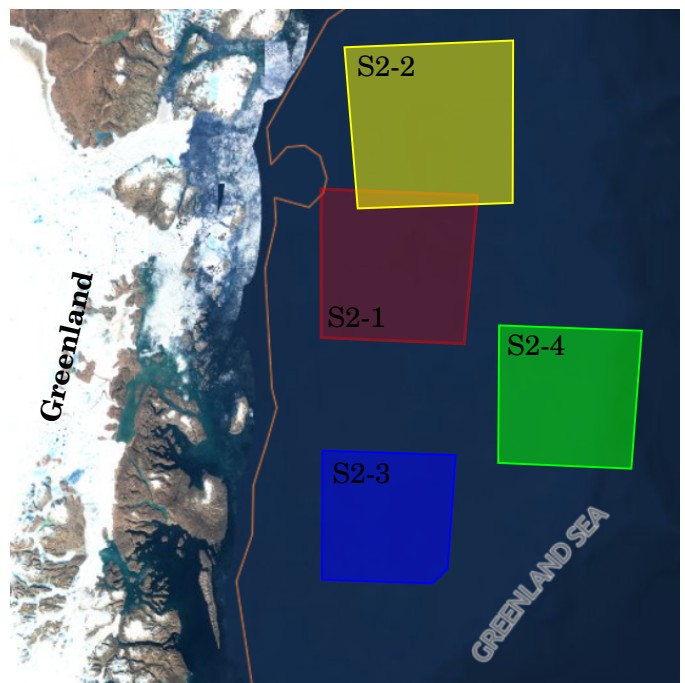

**Figure 2.** Location map of the four Sentinel-2 image data.

## 3 Methodology

### 3.1 Data preparation

By studying various floe images, we find that the intensities of the boundary pixels between two adjacent floes are usually significant higher than those of water pixels, and close to ice pixel intensities. If we consider only two classes in floe segmentation, i.e., ice floe and water, and categorize floe boundary pixels into the class of water, the gap between the two classes will be narrow, and the boundary pixels between two adjacent floes will most likely be classified as ice pixels, with the result that the adjacent floes connect to each other. Therefore, we add an additional class of floe boundary, turning the two-class segmentation task into three-class segmentation problem. In this way, the discriminative ability of the network for segmenting individual ice floes will be improved, not only by broadening the gap between the classes of ice and water but also through the learning of shape and spatial relationships between ice floes and their boundaries.

We also notice that the number of pixels belonging to floe boundary class might be relatively small, comparing with those belonging to the other two classes. Floe boundary is a hard-to-train class with pixel intensities similar to those of ice floe. It is necessary to increase the proportion of floe boundary in the training dataset to balance classes. MIZ images often contain large numbers of small ice floes crowded together, and the proportions of floe boundary pixels in the MIZ images are usually higher



than that of other floe images. Using MIZ images as training dataset is therefore beneficial to reduce the risk of class imbalance in the dataset. Due to this reason, we use MIZ images and their annotations to train DL models for floe instance segmentation.

### 3.1.1 Manual-free data annotation

Manually labeling ice floes in MIZ images is a heavy workload. Efforts are needed to achieve automatic labelling. Because the GVF snake-based method has a superior ability of segmenting large number of small and crowded floes in MIZ images, it is thus utilised as an "annotation tool" to help automatically label individual ice floes in MIZ images. This method first uses the distance map and regional maxima of the binarised MIZ image to automatically locate the initial contours, each of which is a starting set of snake points for the evolutions. Then the GVF snake is run on each initial contour to find floe boundaries. Finally, superimposing all the detected boundaries over the binarised MIZ image, the connected floes are separated and individual floes are determined (Zhang, 2020; Zhang and Skjetne, 2018).

The output of the GVF snake-based method is a binary image with two classes: ice floe and water, as seen in Fig. 3(b). Additional floe boundary class is added by tracing the contours of each segmented ice floes. To enhance floe boundaries and mitigate the imbalance of classes that may still be latent in the training dataset, we widen floe boundaries from one pixel to two pixels in the labels. And with the help of the double thick boundary labels, the network would be able to capture floe boundaries more accurately.

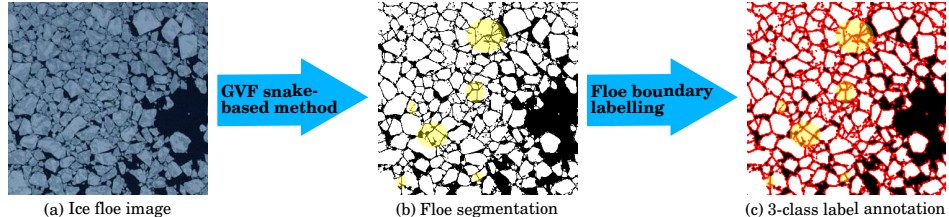

(a) Ice floe image        (b) Floe segmentation        (c) 3-class label annotation

**Figure 3.** Ice floe image annotation. (a) A small MIZ image sample; (b) Floe segmentation of (a) by the GVF-snake based method; (c) 3-class label annotation, produced by adding an additional floe boundary class to (b). Labels: white – ice floe; red – floe boundaries; black – water. Note, the labels annotated by the GVF snake-based method are not always accurate, as seen the highlighted yellow regions in (b) and (c) for instance. Our approach can thus be thought of as self-supervised and weakly-supervised learning that the model will learn from the inaccurate labels created from the data itself without human annotation.

### 3.1.2 Multi-scale division

Our MIZ images are of different sizes. Although the fully convolutional network (FCN) architectures (Long et al., 2015) can be designed for variable-size inputs (Long et al., 2015), in order to reduce memory usage and improve training efficiency, it is more practical to fix the input size of a network to a small value during the training process. Therefore, each of the images, together with the corresponding annotation, is divided into several equal-sized patches that are as close as possible to the fixed input size of the network for training.



Due to the nature of the MIZ in this example, most floes in our image data have similar sizes and shapes. But in more general cases, floe sizes and shapes can vary greatly in an image. Small floes are typically more challenging to segment, not only because of low resolution and small size, but also due to the lack of representation of small objects in training data (Kisantal et al., 2019). To overcome this issue, we select a few floe images and divide each of these images (and also their annotations) randomly into, taking Fig. 4 as an example, $1 \times 1$, $1 \times 2$, $2 \times 3$, $3 \times 4$, $4 \times 5$ sub-images. Note that some sub-images (e.g, $4 \times 5$ sub-images in Fig. 4) may duplicate with the patches. These duplicate sub-images should be removed so that the remaining sub-images, which we refer them as "multi-scale sub-images", and patches appear only once in the dataset, thus preventing data leakage during the training. After automatically removing the sub-images that are duplicated with the patches by, e.g., using a naming convention to overwrite the duplicated ones, the resulted multi-scale sub-images together with the patches constitute our dataset, and they will further be rescaled into the small size required by the network for training. In this way, an ice floe can be resized into several smaller ones of different scales. This will increase the diversity of floe size/shape and the appearance rate of small floes in our training dataset, and thereby helping the network improve the segmentation of small floes.

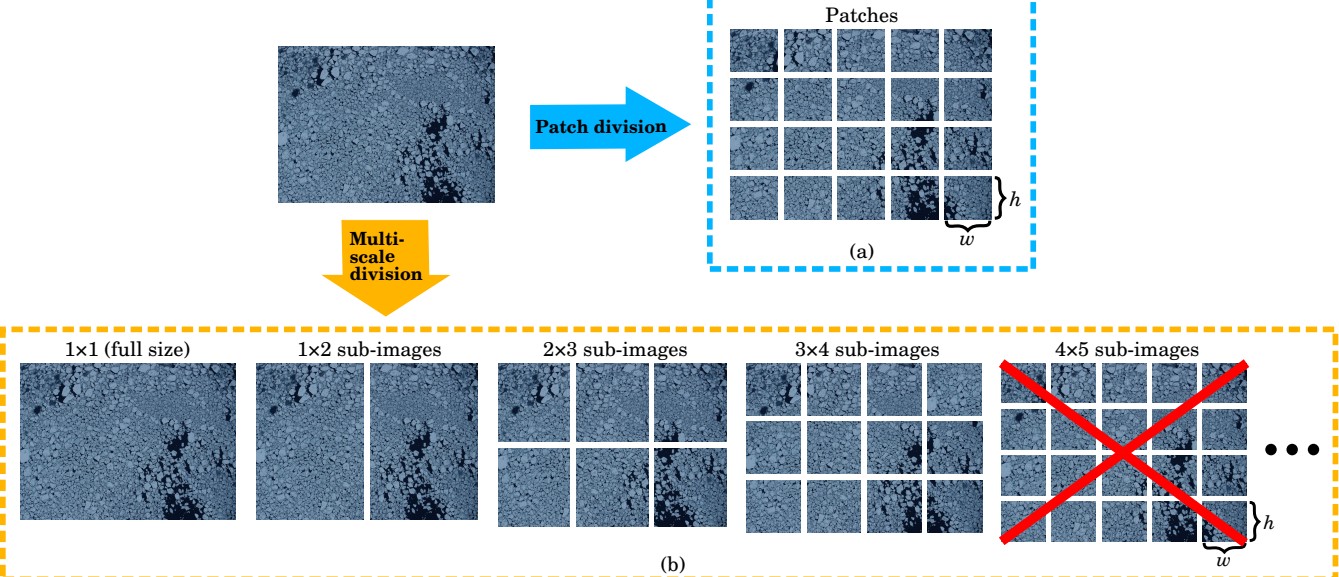

**Figure 4.** Patch and multi-scale divisions. An ice floe image is divided into (a) equal-sized patches with sizes close to network's fixed input size for training, and (b) $1 \times 1$, $1 \times 2$, $2 \times 3$, $3 \times 4$, $4 \times 5$, $\cdots$ sub-images, and the sub-images that duplicate with the patches are removed (we refer to the remaining sub-images as multi-scale sub-images).



## 3.2 Deep learning model

In this research, we use DL semantic segmentation to solve floe instance segmentation problem. We have conducted experiments on training and testing several popular semantic segmentation models on our dataset. By evaluating their performance, we find U-Net++ with the depth of 5 (Zhou et al., 2018) gives the best results, details can be found in Section 5.1.

U-Net++ is an encoder-decoder U-shape symmetric architectures with additional skip connections (Ronneberger et al., 2015). The encoder part encodes the input image into feature maps at multiple different levels, while the decoder part restores the spatial resolution of the feature maps. The encoder and decoder are connected through a series of nested dense convolutional blocks on the skip pathways. U-Net++ also has deep supervision that allows us to use only a single loss layer to determine the optimal depth of the network (Zhou et al., 2018).

## 3.3 Post-processing

Inevitably, a few connected floes may still not be separated and some pixels inside a segmented floe may also be misclassified as edge or water by the DL model, as seen in Fig. 6(b). A post-processing is thus necessary to refine the segmentation made by DL models.

To find potential connected floes/regions, we first calculate the solidities of each segmented floe/region, given by:

$$Solidity(f) = \frac{Area[Filled(f)]}{Area[ConvexHull(f)]} \tag{1}$$

where $f$ is a segmented ice floe/region, $Filled(f)$ is the region of $f$ with all the holes filled in, $ConvexHull(f)$ is the smallest convex polygon that encloses the region of $f$, and $Area[\cdot]$ denotes the area (number of pixels) of the region. The solidity of any segmented floe/region is the ratio of its hole-filled area to its convex hull's area, it reflects the convexity of each segmented floe/region.

An under-segmented region normally has low convexity. That means, a segmented region with low solidity (i.e., the solidity is less than a cut-off threshold $T_s$) and large area (i.e., the area is lager than a threshold $T_a$) is more likely to be composed of connected ice floes. Therefore, we perform the morphological opening (which removes small protrusions and breaks the tenuous connections between objects) to those potential under-segmented regions to detach the floes that are possibly connected. After that, we perform the morphological closing (which fills long thin channels in the interior or at the boundaries of the object) and hole filling to large-area ice floes (i.e., the floes whose area is lager than a threshold $T_a^{'}$), one by one, in order of increasing size.

The procedure of our post-processing algorithm is illustrated in Fig. 5, and Fig. 6 presents the post-processing result of an ice floe image segmentation. Note, a small amount of pixels may be turned to other classes after post-processing. In particular, some isolated ice pixels or tiny floes may vanish during the opening or closing process. To avoid the loss of ice pixels and the underestimation of SIC, the vanished ice pixels are recorded and labelled as boundary pixels in our post-processing algorithm.





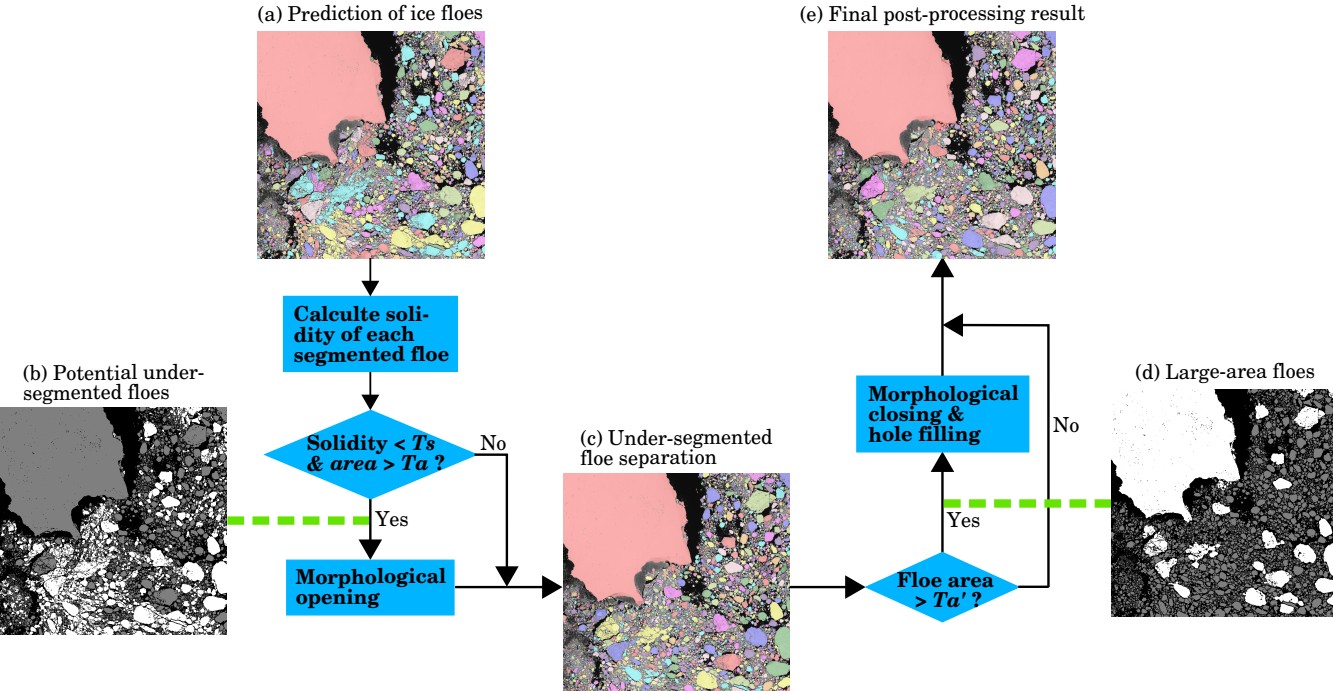

**Figure 5.** Post-processing procedure. (a) Preliminary floe segmentation made by U-Net++; (b) Potential under-segmented ice floes (highlighted in white), obtained by finding the floes whose solidities are lower than a cut-off threshold $T_s$ (which is 0.85 in our case) and areas are larger than a threshold $T_a$; (c) Floe segmentation after performing morphological opening on the floes found in (b), the under-segmented floes are detached; (d) Finding out the floes in (c) whose areas are larger than a threshold $T_a^{'}$ (highlighted in white); (e) Final post-processing result by performing morphological closing and hole filling to the ice floes in (d) one by one in order of increasing size. The segmented floes/regions in (a), (c), and (d) are labelled in different colours. Note that, only ice floes, without any floe boundaries, are presented in this figure.

## 4 Implementation

### 4.1 Training

Our MIZ images were fed into the GVF snake-based method to "label" individual ice floes and floe boundaries. The entire
image processing was fully automatic, using the default parameters in the GVF snake-based method (Zhang, 2018) without any manual tuning. This means that our approach can be thought of as a self-supervised learning, with the supervisory labels being created from the data itself without human annotation (Jing and Tian, 2021).

The processed image pairs, i.e. MIZ images and their annotations, were then divided into the patches with the sizes close to the fixed input size of the network for training, which was 256×256 pixels in our case, a restriction of the amount of MIZ
images and GPU memory. The patch pairs that had severely wrong annotations were removed from the patch dataset. We also chose a few well processed image pairs and divided each of them into several multi-scale sub-images, which were the



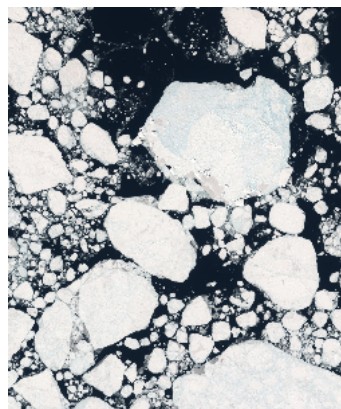

(a) A small subset of S2-1 image.

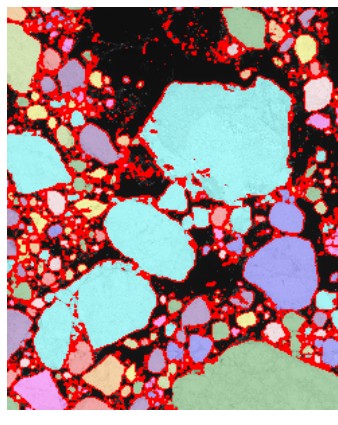

(b) Preliminary segmentation of (a) by U-Net++.

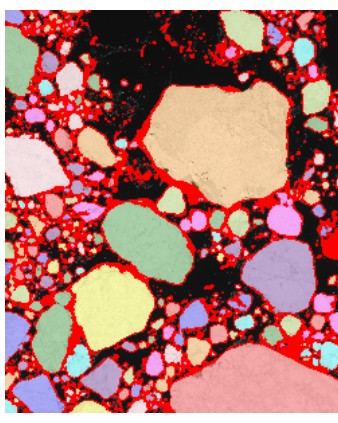

(c) Post-processing of (b).

**Figure 6.** Post-processing. (a) A small subset of S2-1 image; (b) Preliminary segmentation of (a) by U-Net++, some floes are still touching, and some pixel insides a floe are misclassified as edge pixels; (c) Final segmentation after performing post-processing to (b). The detected individual ice floes are labelled in different colours, and the detected floe edges are plot in dark red.

sub-images that did not duplicate with the patches. These patches (333 pairs) and multi-scale sub-images (46 pairs), a total of 379 pairs of images and the corresponding annotations, constituted the final dataset we needed for training a model. And the proportions of each class in the dataset are $41.63\%$ ice floe, $39.83\%$ floe boundaries and $18.54\%$ water.

After randomly splitting the dataset into training (290 pairs), validation (52 pairs), and test (37 pairs) sets, the training and validation dataset were further resized to match the training input size of the network. To increase the amount of training data, data augmentation, including Gaussian blur, rotate, shift, flip, zoom, etc. was also employed in the training. The procedure of our training process is illustrated in Fig. 7.

    It should be noted that, because of the incorrect segmentation made by the GVF snake-based method, some errors, i.e. 180  some labelled ice floes are under- or over-segmented (e.g., the highligted yellow regions in Fig. 3), existed in our dataset, and these errors would reduce the robustness of the trained model. However, we kept the minor erroneous segmentations in our training dataset, leaving the issue to be further overcome by the DL network. Therefore, our approach can also be thought of as weakly-supervised learning that the model will learn from the inaccurate labels (Zhou, 2017). But for evaluating and comparing different DL models, we have manually corrected the errors in our test dataset.

The training was performed on an NVIDIA Tesla P100-PCIE GPU with 12 GB of memory, using TensorFlow 2.4.0. Adam optimizer was applied in the training with a batch size of 4. The learning rate was set to $10^{-4}$ for the first 50 epochs and then to $10^{-5}$ for the next 30 epochs. Note that the proportion of floe boundaries in our dataset is close to that of ice floes, while the proportion of water is a bit smaller than that of the other two classes. The smaller proportion of water has little effect on the model learning to classify water pixels, since water is an easy-to-train class with significantly lower pixel intensities than the



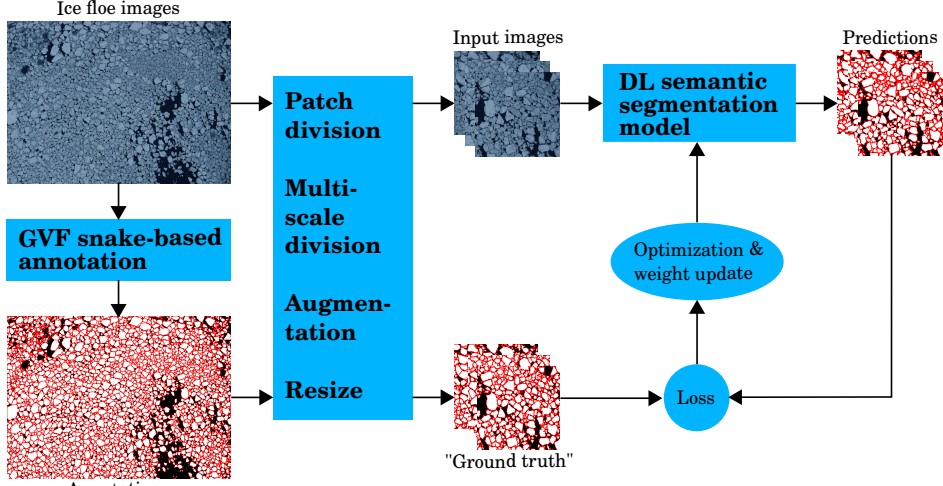

**Figure 7.** Training procedure. Labels in the annotation: white – ice floe; red – floe boundaries; black – water.

other two classes. Therefore, there is no need to modify the loss function and categorical cross-entropy was chosen as the loss in the training.

## 4.2 Inference

Due to the limitation of computer performance, an overall large-scale image was first divided into several small tiles. Each tile was then fed to the trained model to obtain its predicted mask. Thereafter, all the predicted masks were stitched together 195 to restore the spatial pattern, resulting in the preliminary floe instance segmentation. After post-processing the preliminary segmentation, we obtained the final floe instance segmentation. Fig. 8 illustrates the workflow of the inference using the S2-1 image as an example.

## 5 Experimental results and discussions

### 5.1 DL model evaluation

#### 5.1.1 Evaluation metrics

To assess DL model performance, we use accuracy, F1-score (Goutte and Gaussier, 2005), and mean intersection of union mIoU (Long et al., 2015) as evaluation metrics:

$$\text{Accuracy} = \frac{TP + TN}{TP + TN + FP + FN} \tag{2}$$

$$\text{F1-score} = \frac{2TP}{2TP + FP + FN} \tag{3}$$



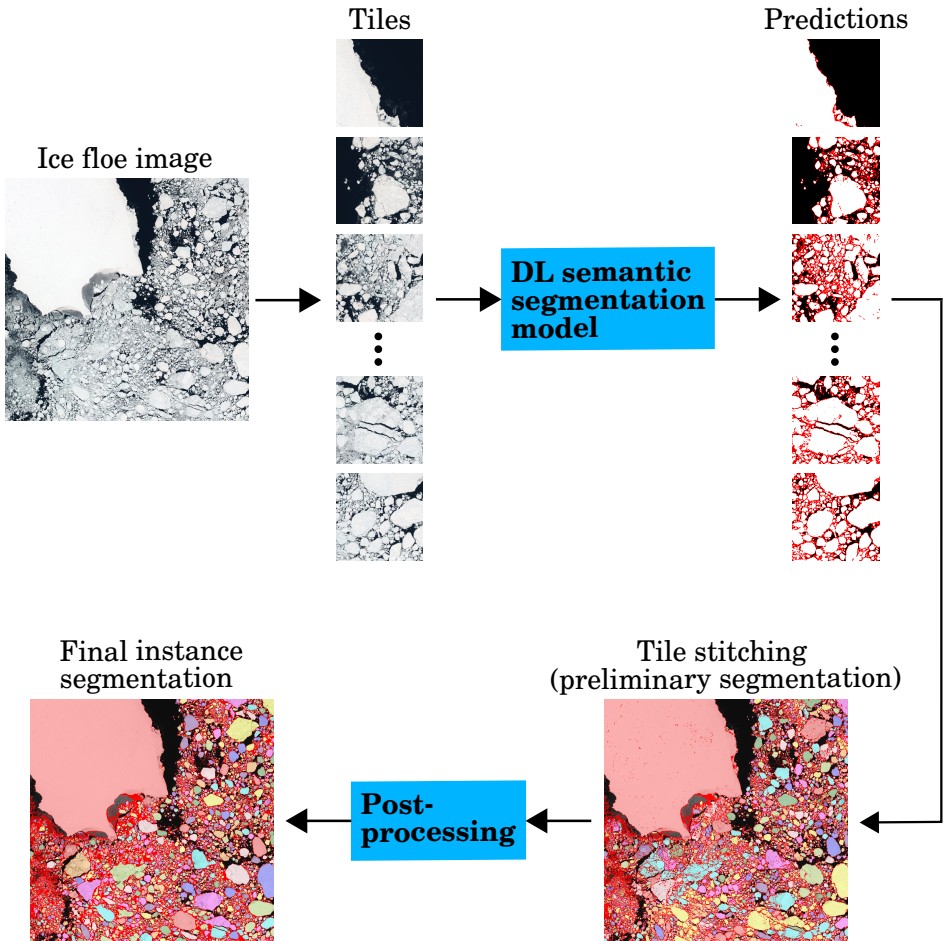

**Figure 8.** Inference procedure of floe instance segmentation for a large-scale image.

$$\text{mIoU} = \frac{1}{C} \sum_c IoU_c = \frac{1}{C} \sum_c \frac{TP_c}{TP_c + FP_c + FN_c} \tag{4}$$

where $TP$, $TN$, $FP$, and $FN$ are pixel-wise true positive, true negative, false positive, and false negative respectively, $C$ is the number of classes, and $IoU_c$ is the intersection of union for class $c$.

With deep supervision, we observed that U-Net++ with the depth of 5 achieved the best floe instance segmentation. We have also conducted experiments to compare the performance of U-Net++ with other SoA semantic segmentation architectures: the family of FCNs (i.e., FCN-8s, FCN-16s, FCN-32s) (Long et al., 2015), SegNet (Badrinarayanan et al., 2017), U-Net (Ronneberger et al., 2015), residual U-Net (ResUNet) (Zhang et al., 2018), and residual U-Net++ (ResUNet++) (Jha et al., 2019), where the optimal depths of U-Net, ResUNet, and ResUNet++ are 5, 5, and 6, respectively. Table 3 lists the evaluation

and comparison results based on our test dataset. We can see that the performance indicators of U-Net, ResUNet, U-Net++



and ResUNet++ are similar to each other and are significantly higher than those of FCN family models and SegNet, while the U-Net++ model gives a slight advantage with the highest scores on our test dataset.

**Table 3.** Model performance indicators.

| Methods | Accuracy (%) | F1-score (%) | mIoU (%) |
|---------|--------------|--------------|----------|
| FCN-32s | 59.52 | 41.05 | 27.96 |
| FCN-16s | 64.92 | 43.04 | 36.33 |
| FCN-8s | 72.66 | 47.00 | 46.21 |
| SegNet | 69.72 | 44.34 | 43.15 |
| U-Net | 83.21 | 49.78 | 62.24 |
| ResUNet | 83.57 | 49.74 | 62.92 |
| **U-Net++** | **83.65** | **49.80** | **63.35** |
| ResUNet++ | 82.81 | 49.74 | 61.21 |

### 5.1.2 Segmentation visualisation

To further investigate the effectiveness of the models, Figures 9 and 10 visualise the floe instance segmentation results on local-scale airborne and global-scale satellite images by U-net, ResUNet, U-net++, and ResUNet++ respectively for comparison.

From these figures, we find that, U-Net is more sensitive to the rapid change in image brightness between neighbouring pixels and detects more noise than others. Although the noise detected by U-Net may help for reducing under-segmentation, it increases the risk of over-segmentation which is difficult to alleviate with post-processing, as seen in Fig. 10(c). On the other hand, ResUNet and ResUNet++ produce less over-segmentation, at the expensive of under-segmentation which results in floes that may not always be effectively detached.

### 5.1.3 Inference time

Table 4 lists the average inference times of different DL models on our test set when using Intel(R) Core(TM) i7-4600U CPU @ 2.10GHz, 16 GB RAM, Integrated Graphics Card. It typically needs more inference time as model complexity increases. FCN-8s and FCN-16s took the least inference time, while ResUNet, U-Net++ and ResUNet++ needed similar inference time that were only 0.4-2 seconds more than other models. The differences in inference time among different DL models were less than an order of magnitude, and could be negligible considering the accuracy differences of the models.

As a compromise, we conclude that the U-Net++ is the most efficient among the other SoA models and is chosen as the DL model for floe instance segmentation.





(a) An airborne MIZ image sample.

(b) Ground truth.

(c) U-Net.

(d) U-Net & post-processing.

(e) ResUNet.

(f) ResUNet & post-processing.

(g) U-Net++.

(h) U-Net++ & post-processing.

(i) ResUNet++.

(j) ResUNet++ & post-processing.

**Figure 9.** Visualisation of MIZ image segmentation results by different models. (a) An airborne MIZ image sample; (b) Ground truth, produced by the GVF snake-based method with manual correction; (c) Segmentation by U-Net; (d) Post-processing of (c); (e) Segmentation by ResUNet; (f) Post-processing of (e); (g) Segmentation by U-Net++; (h) Post-processing of (g); (i) Segmentation by ResUNet++; (j) Post-processing of (i).



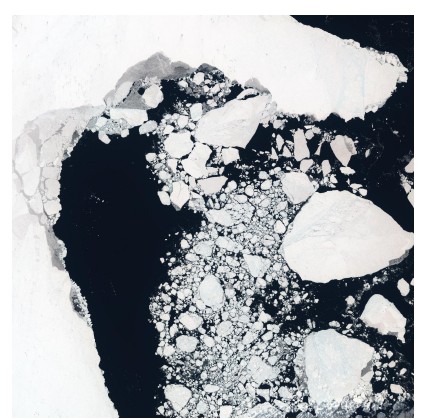

(a) S2-2 image.

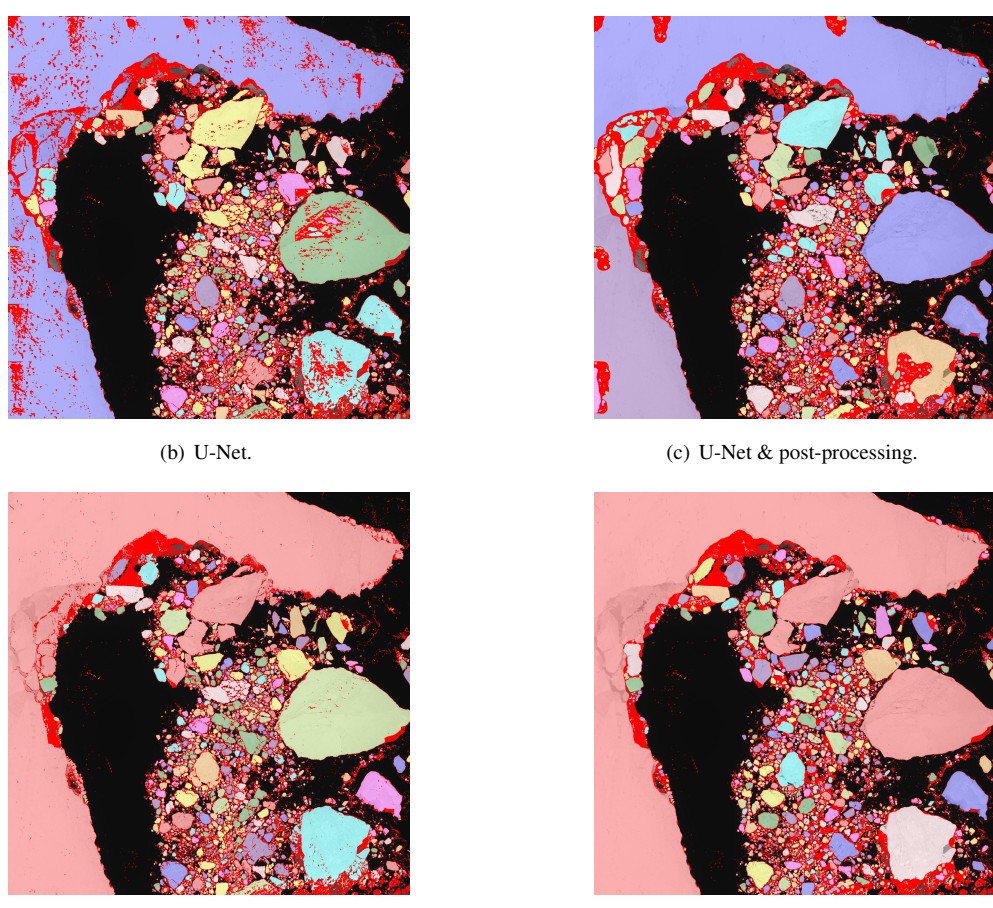

(b) U-Net.

(c) U-Net & post-processing.

(d) ResUNet.

(e) ResUNet & post-processing.

**Figure 10.** Visualisation of S2-2 image segmentation results by different models Part 1. (a) Segmentation by U-Net; (b) Post-processing of (a); (c) Segmentation by ResUNet; (d) Post-processing of (c).





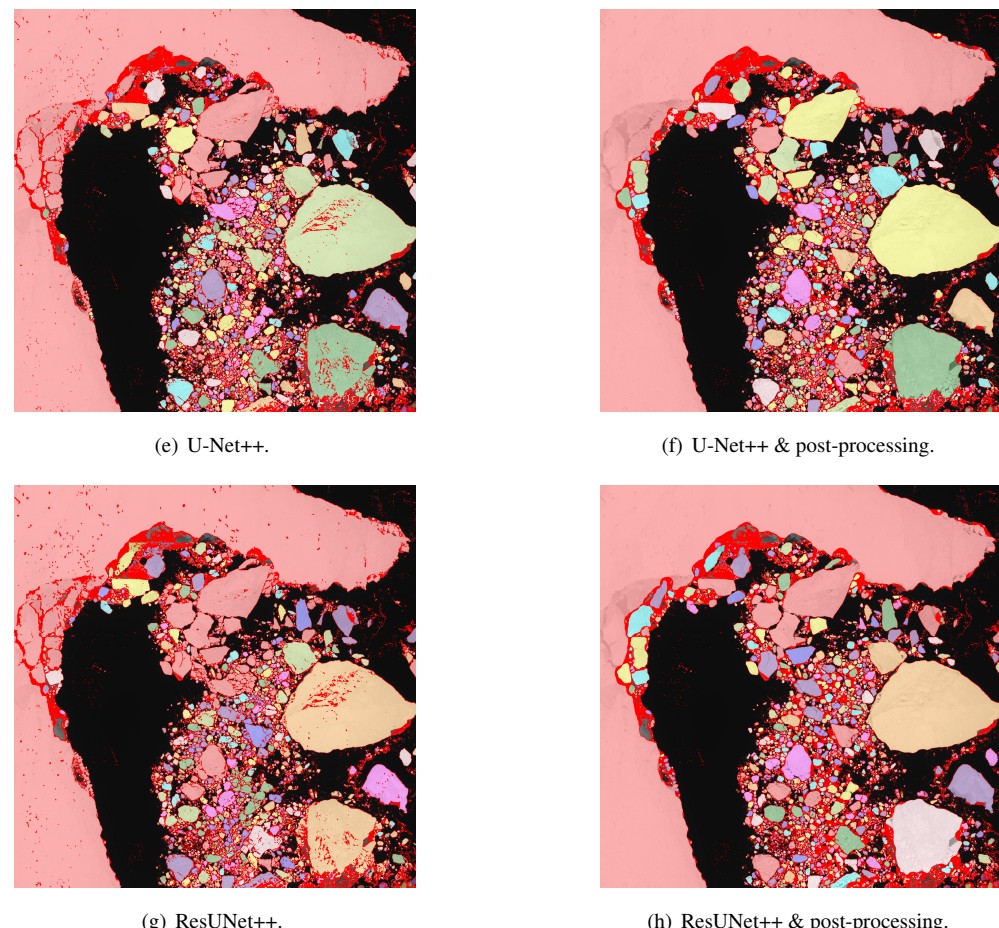

(e) U-Net++.

(f) U-Net++ & post-processing.

(g) ResUNet++.

(h) ResUNet++ & post-processing.

**Figure 10.** Visualisation of S2-2 image segmentation results by different models Part 2. (e) Segmentation by U-Net++; (f) Post-processing of (e); (g) Segmentation by ResUNet++; (h) Post-processing of (g).

## 5.2 GVF-snake based method versus U-Net++

### 5.2.1 Segmentation

Fig. 11 shows an example of floe segmentation on a 256×256 pixel airborne image with about 177 floes processed by the GVF snake-based method and the U-Net++ model-based approach. In this example, the GVF snake-based method over-segmented 9 ice floes and under-segmented 4 floes, while the U-Net++ model under-segmented 45 floes. The U-Net++ model is less sensitive to the weak edges, and also noise, than the GVF snake-based method, and it is more likely to under-segment ice floes

compared with the GVF snake-based method. However, the slight under-segmentation caused by the U-Net++ model can be significantly alleviated by our post-processing method, e.g., leaving only 6 under-segmented floes as seen in Fig. 11(d), while the over-segmentation made by the GVF snake-based method unfortunately cannot be recovered.



**Table 4.** Inference time by using CPU.

| Methods | Without post-processing | With post-processing |
|---|---|---|
| FCN-32s | 0.335 s | 0.389 s |
| FCN-16s | 0.177 s | 0.316 s |
| FCN-8s | 0.124 s | 0.500 s |
| SegNet | 0.957 s | 1.359 s |
| U-Net | 1.220 s | 1.818 s |
| ResUNet | 1.678 s | 2.330 s |
| U-Net++ | 1.674 s | 2.328 s |
| ResUNet++ | 1.682 s | 2.346 s |

Furthermore, due to the nature of the GVF force field, the GVF snake-based method may have difficulty in accurately segmenting both large and small floes when the floes in an image vary greatly in size/shape. Taking Fig. 12(b) as an example, the GVF snake-based method worked well in segmenting most small ice floes, but it failed in segmenting the largest ice floe in the image, dividing the floe into several pieces. This shortcoming of the GVF snake-based method limits its applications to images other than MIZ images, i.e., the images with floes that are of great diversity in size or shape. As shown in Fig. 13, large floes in S2-2 image were severely over-segmented the GVF snake-based method.

Although the U-Net++ model was trained on the pseudo ground truth "annotated" on local-scale MIZ images by the GVF snake-based method, it suffers less from such floe size/shape issues, as seen in Fig. 12(c) in which both large and small ice floes were well segmented by the U-Net++ model. It can also be extended to process HRO satellite imagery data at global scales as seen in Fig. 10(f). The the U-Net++ model-based approach is thus more robust than the GVF snake-based method.

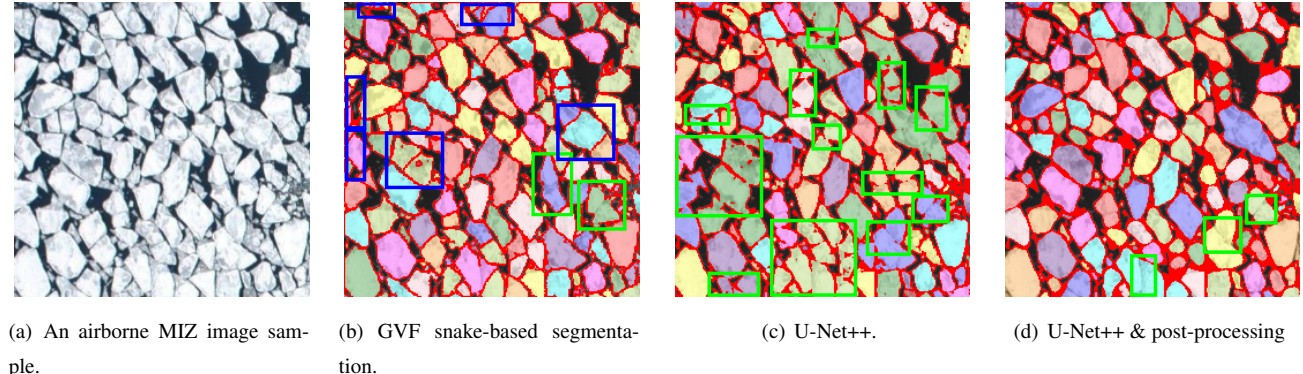

(a) An airborne MIZ image sample.

(b) GVF snake-based segmentation.

(c) U-Net++.

(d) U-Net++ & post-processing

**Figure 11.** Comparison between the GVF snake-based method and U-Net++ on an airborne MIZ image. (a) An airborne MIZ image sample; (b) Segmentation by the GVF snake-based method; (c) Preliminary segmentation by U-net++; (d) Final segmentation after post-processing to (c). Blue rectangle – over-segmentation region; green rectangle – under-segmentation region.





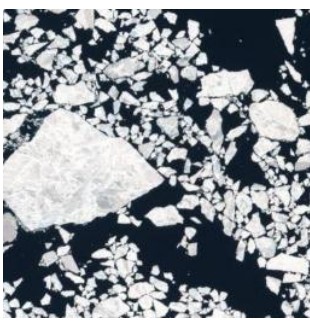

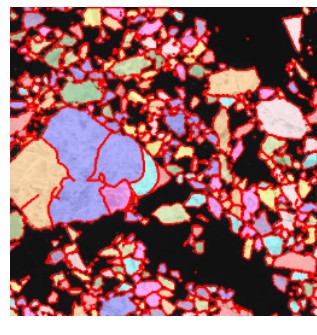

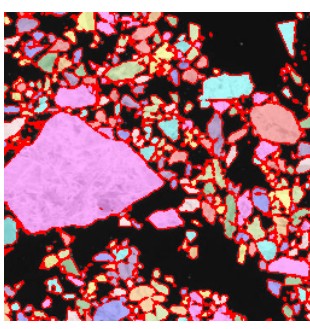

(a) A small subset of S2-4 image.    (b) GVF snake-based segmentation.    (c) U-Net++.

**Figure 12.** Comparison between the GVF snake-based method and U-Net++ on an image containing floes that vary greatly in size and shape. (a) A very small subset of S2-4 image where floes are of various sizes and shapes; (b) Over-segmentation by the GVF snake-based method; (c) Preliminary segmentation by U-Net++.

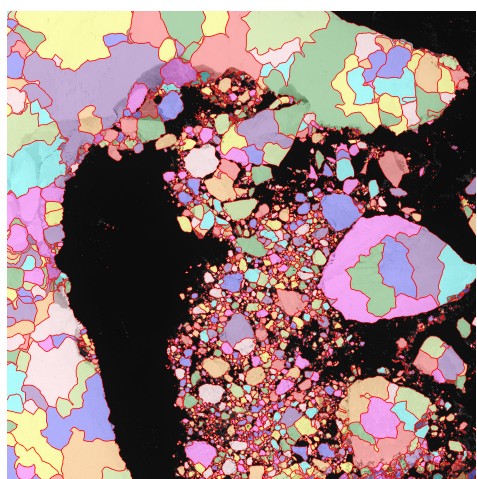

**Figure 13.** Poor segmentation result of S2-2 image produced by the GVF snake-based method.

### 5.2.2 Execution time

The GVF snake-based is a serial approach that detects ice floes one-by-one, and it will take a long time to process an image
with a large amount of ice floes; whereas the U-Net++ model segments individual floes in a parallel manner, meaning that it
will use much less execution time. The difference in execution time between the GVF snake-based method and the U-Net++
model-based approach will increase with larger images or the number of floes presented in the image. For example, with the
CPU with the configurations illustrated in Section 5.1.3, the GVF snake-based method required around 308.263 seconds to find
all the individual ice floes for Figure 11(a), and 60,292.4 seconds to generate a poor floe segmentation from the S2-2 image. In
contrast, the U-Net++ model-based approach took only about 4.234 seconds and 291.050 seconds to complete the segmentation



on these two images with acceptable floe identification results. Table 5 lists the execution times of the GVF snake-based method and the U-Net++ model-based approach on S2 images. It is obvious that the U-Net++ model-based approach is two to three orders of magnitude faster than the GVF snake-based method and is more computationally efficient.

**Table 5.** Execution time by using CPU.

|  | **S2-1** | **S2-2** | **S2-3** | **S2-4** |
|---|---|---|---|---|
| GVF snake-based | 121,490 s | 60,292.4 s | 116,919 s | 139,258 s |
| U-Net++ model-based | 485.52 s | 291.04 s | 540.93 s | 350.23 s |

## 6 Case study: floe size distribution

Our approach has been applied to the airborne MIZ images to segment individual ice floes at local scales, and then extended to process HRO satellite imagery data at global scales, as shown in Fig. 14(b), which exemplifies the detailed floe segmentation result on S2-3 image. With the floe segmentation, it becomes easier to determine SIC and floe characteristics from the image.

Following the existing studies (Rothrock and Thorndike, 1984; Lu et al., 2008), we use the mean caliper diameter (MCD) as the measure of floe size. The MCD of an ice floe is the average over all angles of the distance between two parallel lines, or

270 calipers, that are set against the floe's side walls. It can be simply calculated by (Toyota et al., 2011):

$$d_i = 1.087\sqrt{\frac{4A_i}{\pi}} \tag{5}$$

where $A_i$ is the area of floe $i$. Many studies have shown that the FSD revealed from aerial or satellite images is basically scale invariant, and the cumulative floe number distribution (CFND), $N_c(d)$, which is the number of floes per unit area with MCD no less than $d$, can be represented by a power law function, that is:

$$N_c(d) = \frac{N(>d)}{N_{total}} \propto d^{-\alpha} \tag{6}$$

where $N_{total}$ is the total number of ice floes, and $\alpha$ is the power law exponent used to characterise FSD (Rothrock and Thorndike, 1984; Mellor, 1986; Holt and Martin, 2001; Toyota and Enomoto, 2002; Steer et al., 2008; Lu et al., 2008; Toyota et al., 2011; Perovich and Jones, 2014). In this case study, we follow these studies and use power law exponents that fit the CFND curve to characterise FSD. Fig. 15 shows the FSD determination result based on S2-3 image segmentation, where $\alpha$,

the slope of the power law curve in logarithmic space, was estimated to be 1.52 and 2.04 for floe size smaller and larger than 1000 m, respectively. More FSD characterisation results for local-scale airborne MIZ images and global-scale S2 images can be found in Figs. 16 and 17, respectively.





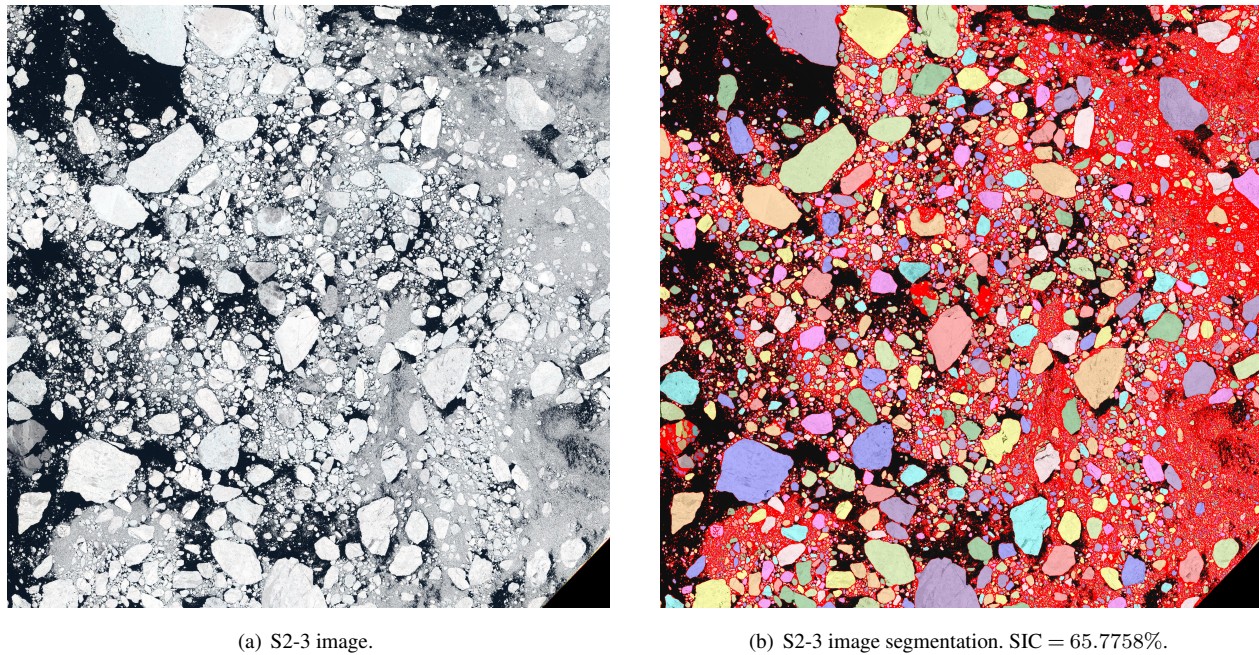

(a) S2-3 image.

(b) S2-3 image segmentation. SIC = 65.7758%.

**Figure 14.** S2-3 image segmentation result. (a) S2-3 image; (b) Floe segmentation of S2-3 image by our approach

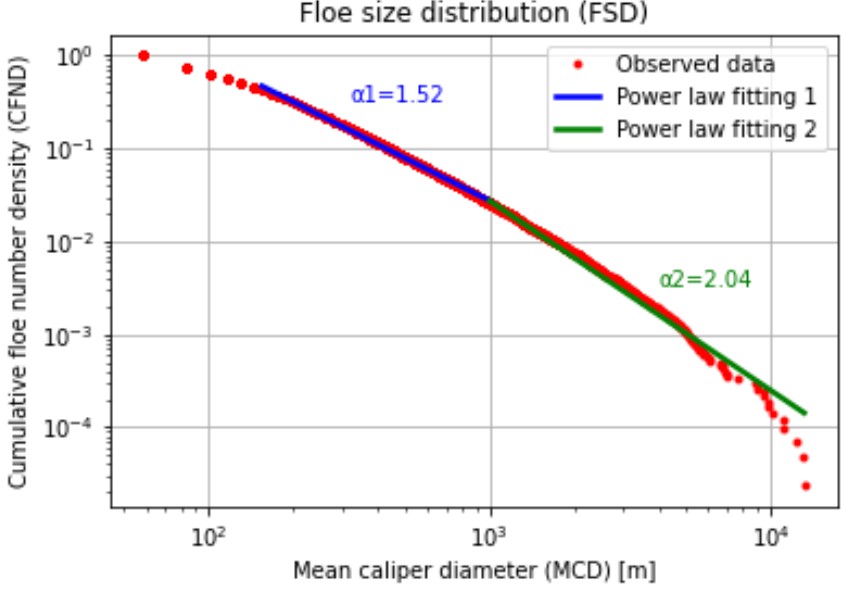

**Figure 15.** CFND determined from S2-3 image.



(a) MIZ-1 image.

(b) MIZ-2 image.

(c) MIZ-1 image segmentation. SIC = 63.6546%.

(d) MIZ-2 image segmentation. SIC = 85.5286%.

(e) CFND of MIZ-1 image.

(f) CFND of MIZ-2 image.

**Figure 16.** Determining and characterising FSD from local-scale airborne MIZ images Part 1. (a) Airborne MIZ image MIZ-1 (b) Airborne MIZ image MIZ-2; (c) Floe segmentation of MIZ-1 image; (d) Floe segmentation of MIZ-2 image; (e) FSD of MIZ-1 image; (f) FSD of MIZ-2 image.





(g) MIZ-3 image.

(h) MIZ-4 image.

(i) MIZ-3 image segmentation. SIC = 84.1209%.

(j) MIZ-4 image segmentation. SIC = 73.7770%.

(k) CFND of MIZ-3 image.

(l) CFND of MIZ-4 image.

**Figure 16.** Determining and characterising FSD from local-scale airborne MIZ images Part 2. (g) Airborne MIZ image MIZ-3 (h) Airborne MIZ image MIZ-4; (i) Floe segmentation of MIZ-3 image; (j) Floe segmentation of MIZ-4 image; (k) FSD of MIZ-3 image; (l) FSD of MIZ-4 image.





(a) S2-1 image.

(b) S2-2 image.

(c) S2-4 image.

(d) S2-1 image segmentation. SIC = 86.6861%,

(e) S2-2 image segmentation. SIC = 65.7758%,

(f) S2-4 image segmentation. SIC = 51.5449%,

(g) CFND of S2-1 image.

(h) CFND of S2-2 image.

(i) CFND of S2-4 image.

**Figure 17.** Determining and characterising FSD from global-scale S2 images. (a) S2-1 image (b) S2-2 image; (c) S2-3 image; (d) Floe segmentation of S2-1 image; (e) Floe segmentation of S2-2 image; (f) Floe segmentation of S2-4 image; (g) FSD of S2-1 image; (h) FSD of S2-2 image; (i) FSD of S2-4 image.



# 7 Conclusions

We have achieved automatic labelling of dataset for training a DL model for ice floe instance segmentation. Our approach has been applied to both airborne and satellite sea ice images to determine FSDs from local to global scales and from low to high SIC. Our experiments show that our approach yields superior results in extracting individual ice floes and is proven to be time-efficient and effective.

Our approach can be used to analyse ice images for determining FSDs and other floe characteristics for use in climate, meteorology, environment and marine operations, etc. Moreover, it can also be utilised as a "higher version" of "annotation tool" and produce more "ground truth" from a wide variety of ice image data sources to further train more robust DL models for obtaining more accurate ice parameters from images.

*Author contributions.* Qin Zhang: Conceptualisation, Methodology, Implementation, Writing – Original Draft. Nick Hughes: Project leader, Quality control, Writing – Review & Editing.

*Competing interests.* We declare that we have no known competing financial interests or personal relationships that could have appeared to influence the work reported in this article.

*Acknowledgements.* This research work was supported by funding from the European Union's Horizon 2020 research and innovation programme under grant agreement 825258, the *From Copernicus Big Data to Extreme Earth Analytics* (ExtremeEarth) project. We would also like to thank the Centre for Research-based Innovation (CRI-SFI) *Sustainable Arctic Marine and Coastal Technology* (SAMCoT) research centre funded by the Research Council of Norway (RCN project number 203471) for providing aerial MIZ image data to our work.



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
