# Peer review of "Ice floe segmentation and floe size distribution in airborne and high-resolution optical satellite images: towards an automated labelling deep learning approach"

_EGUsphere, 2023_

## Referee Comment (RC3)

Dear TC editor and authors of the revised manuscript egusphere-2023-295,

The manuscript topic is interesting and ice floe information is useful for multiple purposes. The other two reviewers have already provided good comments to improve the manuscript. I hope my comments will complement the comments of the other reviewers.

As suggested by another reviewer the abstract needs to be updated to include more on the applied method and results provided by the method instead of general level information. The abstract also begins with FSD and in the manuscript FSD is in the section named as "Case study:FSD" and FSD does not appear in the manuscript title. I suggest at least to remove "Case study:" from the section 6 title, also consider including FSD in the manuscript title. FSD is also significantly present in the introduction section.

The amount of data is very restricted, Why only four Sentinel-2 images have been used? There exist a lot of Sentinel-2 data. It should be emphasized that with such limited data sets this is a case study and the results can possibly not be generalized.

A proper cloud mask is required to be able to automatically segment ice floes. As "manual-free" is mentioned in the manuscript title I think cloud masking should be discussed in the manuscript. Does there exist automated methods for reliable cloud masking or at least excluding images with clouds? Give references of possible cloud masking approaches or suggestions for improved automated cloud masking. Could "manual-free" in the title be "automated" instead?

The results and discussion have now been presented in the same section. I suggest to make a separate "Discussion" section or rather combine "Discussion" with the "Conclusions" section which is very short now.

P2 L36 "Copernicus": give a reference

P3 Dataset → Airborne data:
What is the number of airborne images used in this study? What is "a large amount"? Rather give numbers.
Were the images used as long strips, mosaics, or single shots?

P3 Table 1: Give flight altitude(s) and surface area covered, or their ranges, for the images used. These could possibly be included in the table.

P3 L83 "Cophub": Give reference (URL).

P4 Table 2: Include location information and covered area in the table, e.g. by given center latitude and longitude (and covered area e.g. in km2).

P6 L110-112: Hypothesis of improvement by widening the boundaries would require some evidence. Would it be possible to show test results with a small set of imagery and some numeric evidence based on these tests?

P8 S3.2. Deep learning model: Give at least a short description of U-Net++ giving best results or a diagram of the network. Now this subsection is very short and it is essential for the study.

P8 Post-processing: Applying morphological opening and closing seems a bit heuristic to me. Are there any references or if not would it be possible to demonstrate the benefits of the morphological processing? What is the shape and size of the morphological operator (often a disk is used)? Could

this step be included in the ML algorithm somehow, i.e. could the NN learn the post-processing?

P9 Training: GPU memory is referred on line 170 and this information is then given later on page 10. The hardware (and software) used should be given before it is referred. The used HW and SW could e.g. be included in the dataset section and changing the title to something like "Datasets and computational resources". Also include the used SW with reference in the same section. Also mention there that all the execution times given later are given for this specific configuration.

P10 L174: Does this distribution of classes correspond to their distribution in general? Then it can be used in training. What happens if the distribution of classes is balanced (33% of samples for each class) for the training? Does balancing degrade the classification?

P10 L175: The number of test samples is not very large. What is the effect of reducing samples in training and validation data sets and increasing of the test data set? Are these numbers of samples selected based on some kind of performed tests?

P11 Section 5: Would be good to have some kind of related introductory text under "5 Experimental results and discussions" and "5.1. DL model evaluation", now they are empty.

P13 Section 5.1.3. Inference time: Could this be "Segmentation time" instead, it would be more informative. The HW (and SW) used for segmentation could be given already in an earlier section, e.g. jointly with the introduction of data sets.

P15-15 Figure 10: Fig. 10 is nor in two parts and in two pages. Would it be possible to compress a little and make it to fot on one page?

P19 "6 Case study: floe size distribution": I recommend to drop "Case study:" because FSD is the essential parameter to be estimated by the method and it is also in essential role in the abstract and introduction and the whole manuscript is actually a case study because the datasets are quite limited.

Floe size distribution: Now FSD is estimated in two different resolutions (airborne and satellite data). It would be interesting to see how well FSD can be extrapolated from resolution to another (both from larger to smaller and smaller to larger) based on a fit distribution model. This would be very valuable information and this theme could be included in the discussion section.

P24 "Conclusions": This section is very short. Possibly it could be combined with a discussion section. Here could also be some conclusions on how close to automated FSD estimation the proposed method is? Could it be used for operational monitoring and what will still be required before possible. At least cloud masking should be discussed and also the annual period of possible operation (lighting conditions, what is the fraction of cloudless time in suitable lighting conditions in different sea ice covered areas). What are the ways forward in automated ice floe analysis?

Thank You.

---

## Author Comment (AC1)

In this study the authors used a classical image processing technique to label the ice floe samples, and then used these samples for training a deep learning model, which was used for ice floe segmentation. The authors evaluated the algorithm using two types of remote sensing data and compared its accuracy and runtime with other methods. They claimed that this approach can achieve faster processing speed and higher accuracy. Deep learning models have been widely used in remote sensing image processing, but the prerequisite for obtaining ideal accuracy is usually a sufficient amount of training samples. Sample labeling is usually done manually, which often requires a lot oef manpower and time. Using an automatic labeling method to obtain samples has certain advantages.

Although the deep learning method achieved the best results, which was also expected, using an automatic method for labeling a large number of samples and then for training deep learning models is a commonly used approach. The paper did not provide sufficient innovation, whether in terms of methodology or scientific application. It is recommended that the authors focus more on the methodology itself to address the specific technical issues encountered in the ice floe segmentation, rather than simply using samples to train the deep learning models to obtain so-called high accuracy.

We thank for reviewer for careful review and helpful feedback on our manuscript. Please find our responses to your comments below.

**General comments:**

Using simple methods for automatic labeling of samples and applying them to the training of deep learning models is a common practice, and this paper does not provide enough innovation in this regard. Therefore, I believe that the originality of the paper is relatively limited.

Current methods for extracting individual ice floes and determining floe size distributions from images remain stay on classical approach, which usually require a lot of human intervention and distance away from practical needs.

Although DL techniques have been rapidly developed and successfully applied in a wide range of fields, its application in ice floe segmentation is rarely studied. A main reason limiting the application of DL techniques to ice floe segmentation is the difficulty in obtaining labelled data, which is a very challenging or even impossible task even by domain experts in a manual way. Therefore, this manuscript introduces an approach to automatically label ice floe images, and explores the feasibility of using a small number of labeled datasets to training a DL model for ice floe segmentation and whether the model can be generalized to wider variety of ice floe images. We believe our work is meaningful for the further development of DL in ice floe segmentation, as well as for sea ice studies.

As the authors point out, one of the advantages of this method is that it can reduce the running time. Classical methods for sample labeling take a considerable amount of time. As the number of training samples increases with the further application of the model, the

training time of the model will also increase. If we only compare it with classical methods, this method additionally needs the time for model training. Of course, if we only compare the running time, the deep learning model takes less. But what is the practical significance of shortening the time? Can it be used for some near-real-time applications?

Following the steps introduced in the manuscript to use the classic method for labelling data can reduce unnecessary trial and error time as well as human intervention.

The time saved in processing the data can compensate for the time spent on labelling and training as the amount of data to process increases.

The DL-based method is expected to be applied in marine operations in cold regions such as the Arctic and Antarctic, which require online monitoring of ice conditions in real time and rapid extraction of ice properties to improve maritime safety and provide better data for path planning.

What is the difference between results from classical methods and deep learning methods? The training samples of deep learning come from the classification results of classical methods. If there are some errors in the training samples, these errors may also be introduced into the deep learning model. Although the authors believe that deep learning can overcome this problem by itself, the influence will still exist. How do the classical methods and deep learning methods affect the subsequent acquisition of ice floe parameters, and is the difference obvious?

In shortly, the classical method, i.e., GVF snake-based, identifies floe boundaries one by one and takes longer to process images as the number of ice floes increases. The GVF snake-based method also does not well in global-scale floe image segmentation and tends to over-segment big floes.

The DL-based method identifies floe boundaries simultaneously and takes shorter processing time (please see Tab. 5 in the manuscript). Although the DL model was trained on the data annotated by the classical method, it surfers less from the segmentation issue (please see Fig. 12 and 13 in the manuscript, and the figures blow. Please also pay attention to the different colours in the figures which indicate whether the floes are under-/over-segmented). The reason for this, i.e., the explanation of the rest questions in this comment, can be found in our response to your comment "Line 22".

**S2 image**

**GVF** snake-based**

---

## Author Comment (AC2)

Floe size distribution (FSD) becomes a very important parameter in nowadays sea ice modelling; however, high-resolution imagery seems to be the only source to obtain such kind of information. Thus, an automatic image-processing method is also important in this field. This study provided a deep learning-based segmentation method to process airborne and optical satellite images, and obtained good results of FSD. It seems that a completely automatic method to get FSD becomes possible.

Actually, it is not the first time for me to review this manuscript. I understand the solid revision that the authors have conducted to improve the paper. I still encourage the authors to address the remaining issues, and make the manuscript smoother to follow. Such an interesting topic merits publications and will be valuable for more accurate access on FSD.

We thank for reviewer for careful review and helpful feedback on our manuscript. Please find our responses to your comments below.

The abstract talks more about the background, instead of the solid achievements in the present study. I suggest a shorter background, and more results of the present study should be presented.

Thanks for the suggestion! We will revise the abstract.

1. Is there any relationship between the airborne data in 2.1 and the satellite data in 2.2? Or both of them are employed here just to test the effect of the new method on different kind of imagery.

There is no relationship between the airborne data and satellite data.

The local-scale airborne MIZ data were used to train and test DL models. The satellite data were additional data only used to test the generalization ability of the DL models from local to global scale, and they were never encountered by the DL models during the training.

2. Lines 210-215. "U-Net++ with the depth of 5 achieved the best floe instance segmentation", is this a result of "experiments to compare the performance of U-Net++ with other SoA semantic segmentation architectures"? I mean if you have known U-Net++ is the best among all, why do you compare them again? And for the other methods such as ResUNet, ResUNet++, additional explanations should be added here to tell the difference between them.

Yes, it is the result of the comparison among different DL models.

It is common to show performance comparisons between different DL models if using DL method. We will move this model comparison subsection to the appendix and add more descriptions and diagrams about models.

3. There are two fig10e. And for figures 9-10, the difference between these results are very difficult to distinguish if no additional notations such as in fig11e are presented.

Thanks for the suggestion! We will add notations to these figures and correct the wrong figure numbering.

4. It is a little difficult for me to follow the contents in sections 4 and 5. A possible reason is that so many names of processing methods are presented here, and also two kinds imagery are included as examples to show the effect of these methods. I was not told why airborne data were employed here but satellite imagery were employed there. Thus the main improvement of the present study are submerged by these information.

We apologize for the confusion. We will adjust the structure of the manuscript and revise sections 4 and 5.

5. There are some very interesting results in section 6, for the variations in the power-law exponent, can you give some more explanations on them? Otherwise, it is not necessary to present so many pictures as example without any discission.

It is common to use some typical images to show the effect of the method. Therefore, we chose these floe images with low to high ice concentrations to present segmentation results made by the DL-based methods.

The determination of the ice floe size distribution is a further application after the ice floe segmentation, and it is also part of our ongoing project. We thanks for the suggestion, and we will give a brief explanation on it in the revised manuscript.

6. There is a so quick stop in the conclusion section, can you give some evaluations on the limitations of the present study?

Apologies for the short conclusion and thanks for the valuable suggestion! We will add some descriptions of the limitations of this study in the revised manuscript, e.g., floes with many melt ponds and cloud masking.

---

## Author Comment (AC3)

Dear TC editor and authors of the revised manuscript egusphere-2023-295,

The manuscript topic is interesting and ice floe information is useful for multiple purposes. The other two reviewers have already provided good comments to improve the manuscript. I hope my comments will complement the comments of the other reviewers.

We thank for reviewer for careful review and helpful feedback on our manuscript. Please find our responses to your comments below.

As suggested by another reviewer the abstract needs to be updated to include more on the applied method and results provided by the method instead of general level information. The abstract also begins with FSD and in the manuscript FSD is in the section named as "Case study:FSD" and FSD does not appear in the manuscript title. I suggest at least to remove "Case study:" from the section 6 title, also consider including FSD in the manuscript title. FSD is also significantly present in the introduction section.

Thanks for the comments. We will revise the abstract, reorganize manuscript structure, and include ''floe size distribution'' in manuscript title.

The amount of data is very restricted, Why only four Sentinel-2 images have been used? There exist a lot of Sentinel-2 data. It should be emphasized that with such limited data sets this is a case study and the results can possibly not be generalized.

Our approach has been applied to extract the ice floes spatially and temporally from Sentinel-2 images, e.g., below show the floe segmentation results for S2-1 region from April to June in 2021 and other years.

2021

2020                              2019

[Figure]

It is not practical to present so many images in the paper, and instead it is common to use some typical images to demonstrate segmentation performance of the methods. Therefore, in addition to four airborne MIZ images, we also used these four typical Sentinel-2 floe images with ice concentration from low to high to show the floe segmentation results.

Please note that the DL models were trained only on local-scale airborne MIZ images, and they did not encounter global-scale satellite data during the training. The Sentinel-2 images were extra data used to investigate the generalization abilities of the trained DL models from local-scale MIZ images to global-scale images (please see our response to reviewer 1's comment ''Line 98'').

Please note also that, how many individual floes (especially those that tightly connected with many other floes) are successfully identified from an image without being over- or under-segmented is also an important measure of ice floe segmentation method.

A proper cloud mask is required to be able to automatically segment ice floes. As "manual-free" is mentioned in the manuscript title I think cloud masking should be discussed in the manuscript. Does there exist automated methods for reliable cloud masking or at least excluding images with clouds? Give references of possible cloud masking approaches or suggestions for improved automated cloud masking. Could "manual-free" in the title be "automated" instead?

There is an option to choose cloud coverage when downloading Sentinel-2 data from the Copernicus Open Access Hub. So it can exclude images with clouds.

In addition, we have applied and compared the existing methods, Sen2Cor (via ESA Snap software or running on linux terminal) and Fmask, to mask clouds in Sentinel-2 images by means of cloud masking and classification. For us, Fmask works better than Sen2Cor. Please see below for reference.

Thanks for the comment, we will add a brief discussion on cloud masking in the revised manuscript, and also revise manuscript title.

=====

Cloud masking for Sentinel-2 floe image segmentation

Thin Cloud

- Cloud mask: some floes may be mistaken as cloud

- Segmentation: some thin cloud pixels may be classified as ice edges (leading to over-estimation of SIC)

- Some floes under thin cloud can be identified by the model, but they may be masked out by the cloud mask

S2 image

Cloud mask (sen2cor)

Segmentation

Classification (sen2cor)

**Thick Cloud**

- Cloud mask: some cloud regions may not be detected

- Segmentation: it is hard to find floes and floe boundaries in and around the regions covered by thick cloud

Suggestion:
The detected floes adjacent to cloud regions will not be considered when determining FSD, since they are likely to be incorrectly segmented

S2 image

[Figure]

Segmentation

Cloud mask (sen2cor)

[Figure]

Classification (sen2cor)

[Figure]

S2 image

[Figure]

Before cloud masking

[Figure]

After cloud masking

[Figure]

Green: clouds, detected by sen2cor
Blue: floes adjacent to the detected clouds

S2 image

[Figure]

Before cloud masking

[Figure]

After cloud masking

[Figure]

Green: clouds, detected by sen2cor
Blue: floes adjacent to the detected clouds

The current model doesn't consider the category of cloud/cloud shade, some cloud/cloud shade pixels are mistaken for ice edges.
Also due to the insufficient detection of cloud by the sen2cor module, it's hard to correct the this type of error.

**Sen2Cor vs Fmask**
**Via cloud mask - thin cloud**

[Figure]

**Sen2Cor vs Fmask**
**Via cloud mask - thick cloud**

[Figure]

**Sen2Cor vs Fmask**
**Via classification**
**- thin cloud**

[Figure]

[Figure]

The results and discussion have now been presented in the same section. I suggest to make a separate "Discussion" section or rather combine "Discussion" with the "Conclusions" section which is very short now.

Thanks for the suggestion, we will adjust manuscript structure.

P2 L36 "Copernicus": give a reference

The link: https://scihub.copernicus.eu will be added there.

P3 Dataset → Airborne data: What is the number of airborne images used in this study? What is "a large amount"? Rather give numbers. Were the images used as long strips, mosaics, or single shots?

The total number of airborne images we got was 254, of which 52 were selected to be labelled for this study.

The airborne image data were used as single shots.

P3 Table 1: Give flight altitude(s) and surface area covered, or their ranges, for the images used. These could possibly be included in the table.

We apologize that we cannot provide the information you suggested.

Neither of the authors of this manuscript was on board the expedition 8 years ago. The airborne images were kindly provided from other research group at a university. Table 1 and the average resolution for a pixel were the only information we got about the airborne images.

P3 L83 "Cophub": Give reference (URL).

The URL was given in the 1st ref. It will be moved to the end of ''Copernicus Open Access Hub''.

P4 Table 2: Include location information and covered area in the table, e.g. by given center latitude and longitude (and covered area e.g. in km2).

We will add these information to the table.

P6 L110-112: Hypothesis of improvement by widening the boundaries would require some evidence. Would it be possible to show test results with a small set of imagery and some numeric evidence based on these tests?

Yes, we can do that when submitting the revised manuscript.

Please note, the widened boundaries (2-pixel wide) are inner boundary (1-pixel wide) and outer boundary (1-pixel wide), which are the two types of boundaries for an object/floe. Please also see our response to reviewer 1's comment ''Line 111-112''.

P8 S3.2. Deep learning model: Give at least a short description of U-Net++ giving best results or a diagram of the network. Now this subsection is very short and it is essential for the study.

We apologize for the lack of descriptions about the models, and we will add the descriptions and diagram of the models in the revised manuscript.

P8 Post-processing: Applying morphological opening and closing seems a bit heuristic to me. Are there any references or if not would it be possible to demonstrate the benefits of the morphological processing? What is the shape and size of the morphological operator (often a disk is used)? Could this step be included in the ML algorithm somehow, i.e. could the NN learn the post-processing?

Please see (Banfield, 1991; Banfield 30 and Raftery, 1992; Soh et al., 1998; Steer et al., 2008; Wang et al., 2016) for reference. We also mentioned their work in the Introduction section:

''Morphological operations can be used with different improvements to determine individual ice floes, but the methods operate directly on binarized floe images and thus cannot separate out the floes that had no or few gaps with any surrounding floes after binarization (Banfield, 1991; Banfield 30 and Raftery, 1992; Soh et al., 1998; Steer et al., 2008; Wang et al., 2016)''

Due to the problems with the morphological operations described above, we used the morphological operations in a post-processing step to refine the floe segmentation. Fig. 5 and Fig. 6 in the manuscript can demonstrate the benefit of the processing. A disk-shaped

structuring element with a radius of 4 pixels was used in the morphological operations. The disk shape was chosen because it is non-directional and can handle ice floes more uniformly than other shapes without being aware of floe's irregular shape and orientation.

The next task after the work presented in the manuscript is to use the trained DL model together with the post-processing to label more ice floe images and create more dataset, and then train more robust DL models for ice floe segmentation, as we mentioned in the Conclusion section: ''it can also be utilised as a "higher version" of "annotation tool" and produce more "ground truth" from a wide variety of ice image data sources to further train more robust DL models for obtaining more accurate ice parameters from images.''

P9 Training: GPU memory is referred on line 170 and this information is then given later on page 10. The hardware (and software) used should be given before it is referred. The used HW and SW could e.g. be included in the dataset section and changing the title to something like "Datasets and computational resources". Also include the used SW with reference in the same section. Also mention there that all the execution times given later are given for this specific configuration.

Thanks for the suggestion! We will adjust them in the revised manuscript.

P10 L174: Does this distribution of classes correspond to their distribution in general? Then it can be used in training. What happens if the distribution of classes is balanced (33% of samples for each class) for the training? Does balancing degrade the classification?

The distribution of classes depends on the extent of sea ice coverage and the amount of ice floes in the image. The higher the ice concentration in the image, the higher the proportion of ''ice'' class (the lower the proportion of ''water'' class); the more ice floes, the higher the proportion of ''floe boundary'' class. MIZ images generally have higher proportion of ''floe boundary'' class than other ice floe images.

Balanced classes will not degrade the classification. Instead, it makes model easier to train because it helps the model learn the features of each class equally.

In ice floe segmentation, ''floe boundary'' is a hard-to-train class because: 1) the pixel intensity of floe boundaries, especially the boundaries between connected floes, is usually similar to that of ice; 2) the proportion of floe boundaries in ice floe images is usually much smaller than other two classes of "ice" and ''water'' (MIZ images are less affected by this issue). Therefore, it is necessary to increase the proportion of ice floe boundaries in the training data set, and using MIZ images as training images and widening floe boundaries can help with this.

Although the DL model was trained on MIZ images with a relatively high proportion of ice floe boundaries, it also has a good generalization ability to other ice floe images with low proportion of floe boundaries (e.g., the largest ice and water regions in Sentinel-2 images). It

demonstrates the model trained on restricted datasets that can also generalize to wider datasets.

P10 L175: The number of test samples is not very large. What is the effect of reducing samples in training and validation data sets and increasing of the test data set? Are these numbers of samples selected based on some kind of performed tests?

Reducing the training and validation sets to increase the test set often leads to a decrease in the performance of DL models.

A common ratio between train, validation, and test is 80:10:10, and 70:15:15, 60:20:20, etc. are also practical ratios for splitting datasets.

Regarding the number of test samples, please see our response to your previous general comment ''The amount of data is very restricted, …''.

P11 Section 5: Would be good to have some kind of related introductory text under "5 Experimental results and discussions" and "5.1. DL model evaluation", now they are empty.

We apologize for the lack of descriptions, and we will put some descriptions in the revised manuscript.

P13 Section 5.1.3. Inference time: Could this be "Segmentation time" instead, it would be more informative. The HW (and SW) used for segmentation could be given already in an earlier section, e.g. jointly with the introduction of data sets.

Thanks for the suggestion and we will change them.

P15-15 Figure 10: Fig. 10 is nor in two parts and in two pages. Would it be possible to compress a little and make it to fot on one page?

We will put the figures on one page.

P19 "6 Case study: floe size distribution": I recommend to drop "Case study:" because FSD is the essential parameter to be estimated by the method and it is also in essential role in the abstract and introduction and the whole manuscript is actually a case study because the datasets are quite limited.

Floe size distribution: Now FSD is estimated in two different resolutions (airborne and satellite data). It would be interesting to see how well FSD can be extrapolated from resolution to another (both from larger to smaller and smaller to larger) based on a fit distribution model.

This would be very valuable information and this theme could be included in the discussion section.

Thanks for the suggestion! Multi-scale FSD is actually part of our ongoing project. We will try to give a brief explanation on multi-scale FSD in the revised manuscript.

P24 "Conclusions": This section is very short. Possibly it could be combined with a discussion section. Here could also be some conclusions on how close to automated FSD estimation the proposed method is? Could it be used for operational monitoring and what will still be required before possible. At least cloud masking should be discussed and also the annual period of possible operation (lighting conditions, what is the fraction of cloudless time in suitable lighting conditions in different sea ice covered areas). What are the ways forward in automated ice floe analysis?

We apologize for the short conclusion and thanks for the valuable suggestion!

In the revised manuscript we will 1) discuss the cloud masking 2) discuss the limitations of the method 3) give the application prospects of the method 4) give suggestions on the way forward in automated floe analysis.

---

## Author Response (AR2)

Dear Authors

I have received feedback from two reviewers who have reviewed the first and second rounds of your revised manuscript. One reviewer is satisfied with your latest revision. Another reviewer is still sceptical about the novelty of this work. Both reviewers point out some comments that would require the authors to make minor revisions to the manuscript. I accordingly invite you to make further improvements to your manuscript, please pay particular attention to the novelty of your work. Then, I would be willing to accept your manuscript for publication.

Best regards,

Bin Cheng

We are grateful to the editor for the opportunity to improve our manuscript. We have addressed reviewers' comments point by point and revised our manuscript. Please find our responses below.

===========================================================================

Reviewer 1:

After the last round of peer review, the author made significant revisions to the paper, resulting in a noticeable improvement in overall quality. I am generally satisfied with the author's response and revisions. However, I still believe that the paper lacks innovation. While the author argues that their method has some relevance for future research in ice floe segmentation, I find the method itself lacking clear scientific significance and methodological value. The author's substantial work primarily involves labeling samples using existing methods and then training a deep learning model for classification. The improvements to the method itself are limited, and it can be said that the main contribution lies in the application of deep learning methods to ice floe segmentation, although this application does hold a certain level of importance in the field.

We appreciate the reviewer's careful review and valuable comments on our manuscript. With regards to the novelty and innovation of the study, this lies in the application of our floe mapping approach to widely available, and free to access, near-real time (NRT) high resolution optical (HRO) images from Copernicus Sentinel-2. Previous studies, e.g. have been limited to airborne or declassified military satellite images (e.g. MEDEA used in Denton and Timmermans, 2022, and Wang and others, 2023) or X-band SAR (Ren and others, 2015, Hwang and others, 2017). Neither of these can provide the regional, NRT coverage to make them viable in operational monitoring for maritime safety. We appreciate that HRO is still cloud and nighttime darkness limited, but the benefit of using Sentinel-2 is in combination with other approaches, especially SAR-based classifications, when cloud-free periods allow. We look forward to the darkness limitation being solved with the future Copernicus LSTM mission.

Please find our responses to the specific comments below.

I suggest the following improvements to the paper:

1. Currently, the method comparison is placed in the appendix. I recommend moving it to the main body of the paper to make it more accessible for readers to read and understand.

Thank for reviewer's suggestion. We have moved the entire appendix to the main body of the manuscript.

2. The architecture of the DL has been described. Not having deep knowledge when it comes to the design of deep learning networks, I am wondering why exactly this architecture was chosen? Have the authors performed tests on different ones, or is there some logical reasoning behind the number of convolutional and pooling layers, or does the exact architecture not really matter as long as you have "enough" layers?

Yes, we have performed several tests for each DL model on different numbers of convolutional layers, batch normalization, drop out, kernel size, etc. The performances of the same model under different parameter settings did not improve significantly, while the performances of different DL models varied. Since the currently available training datasets are still limited, it is premature to emphasize the exact architecture which may lead readers to mistakenly think that the current model is robust enough for any complex ice conditions. As training datasets become richer, more robust floe segmentation models will be developed. Specifying an exact model architecture then makes more sense.

3. How does the method perform when dealing with unique sea ice conditions, such as the presence of melt ponds? This aspect should be addressed in the paper.

We are continuing to process sea ice images with challenging ice conditions. But an idea following from this is to potentially use additional sea ice classification methods to detect melt pond regions. Then catalog these melt pond pixels to the class of floe in DL prediction, and finally refine the floe segmentation using the proposed post-processing.

Thank you for the comment. We have added this in the revised manuscript.

=====================================================================

Reviewer 2:

The authors have made some solid revisions according to the comments from several reviewers. I think the manuscript has greatly improved and I only have some minor comments this time.

We thank the reviewer for their careful review and valuable comments on our manuscript. Please see our response to your comment below.

1)   (Line 130) It is, obviously, convenient to use GVF snake-based method as an "annotation tool" to help automatically label individual ice floes. However, whether automatically labeling would affects the result?

Automatic data labelling would not affect the results since the GVF snake-based method produces good individual ice floe segmentation results from local-scale MIZ images, allowing DL models to be trained on well-labelled dataset.

2)   (Line 138) The boundary line consists of two pixels: inner and outer boundaries of floes. Does it mean that the inner boundary consist of one ice pixel and the outer boundary consist of another ice pixel?

The inner boundary consists of pixels from the floe itself. While the outer boundary mainly consists of pixels from ambiguous edges between the floe and another floe or the water.

3)   (Line 150) Why "1×1, 1×2, 2×3, 3×4, 4×5 sub-images", not "1×1, 2×2, 3×3, 4×4, 5×5 sub-images"? Or in other words, what's the advantages of the former cropping approach?

An overall image was divided randomly into sub-images. Fig. 4 is just an example to illustrate this multi-scale division process. We apologize for the confusion, and we have modified the description.

4)   (Line 238) Did the validation (52 pairs) and test (37 pairs) are also be trained? As illustrated in the abstract, is it enough to only take 290 pairs into training for successful results?

No. The validation dataset was used to evaluate how well the model fits the training dataset during training, e.g., to check whether the model is underfitting or overfitting the training dataset. It does not directly participate in the update of model weights. While the test dataset was used to evaluate the performance of final trained model, e.g., to compare the performance between different models.

The DL model in this work was trained on 290 pairs training dataset and produces acceptable results. Thanks for the comment. We have modified the wording in the abstract.

5)   (Line 356) How many images the average segmentation time was calculated from, if it could be told? And I think it is also necessary to explain the hardware used. In addition, should training time also be taken into account?

37 test images were used to compare the segmentation time between models. The hardware was Intel(R) Core(TM) i7-4600U CPU $@$ 2.10GHz, 16 GB RAM, Integrated Graphics Card, which was described in Section 2.2 Software and hardware.

The required training time from fast to slow is FCN family, SegNet, U-Net & ResUNet, U-Net++ & ResUNet++. The differences between models are only a few minutes and could be negligible.

6)   (Line 364) It was obvious that there were many over-segmentation on ice floes without post-processing as shown in Fig.A3. Does that mean for now there aren't auto approaches to overcome the over-segmentation good enough? In this way, whether post-processing has become a necessary step in the whole processing progress? What's more, will post-processing affect the automation of whole processing progress?

So far, we have not found a good solution to the over-segmentation problem for a single optical floe image, especially for over-segmentations that are ambiguous (e.g., the middle-left part of the biggest floe in S2-2 image in Fig. A3/Fig. 11). A series of images of the same area over different periods or other types of the remote sensing data may be needed to help determine whether the floe is over-segmented.

Post-processing is used to refine the floe segmentation, such as removing "holes" in large floes, preserving floe shape, and separating potentially under-segmented floes that may affect FSDs. This is a necessary step to maintain the integrity of the ice floe, especially when the surface of the floe is noisy, i.e. contains small melt ponds, partially covered by small clouds, etc.

In our post-processing, we have proposed criteria to automatically check whether the detected floes are well segmented to decide whether to perform subsequent morphological opening and closing. The area cut-off threshold $Ta$ for finding the potential under-segmented floes and the area threshold $Ta'$ for finding floes require a smooth shape are two main parameters in our post-processing that need to be adjusted according to image scale and/or practical application needs, while the solidity cut-off threshold Ts kept constant at 0.85. The pure morphological operations adopted in our post-processing often require extensive manual parameter tuning to segment floes even from a single image, since they operate on the binarized image and relies on how well the edges are detected between connected floes. As the DL method detects more accurate floe boundaries, morphological operations in our post-processing become less dependent on parameters in refining floe segmentation (a disk-shaped structuring element with a radius of 4 pixels was used in the morphological operations), making the entire process more automated.